# Mortality in Severe Antibody Deficiencies Patients during the First Two Years of the COVID-19 Pandemic: Vaccination and Monoclonal Antibodies Efficacy

**DOI:** 10.3390/biomedicines10051026

**Published:** 2022-04-29

**Authors:** Cinzia Milito, Francesco Cinetto, Andrea Palladino, Giulia Garzi, Alessandra Punziano, Gianluca Lagnese, Riccardo Scarpa, Marcello Rattazzi, Anna Maria Pesce, Federica Pulvirenti, Giulia Di Napoli, Giuseppe Spadaro, Rita Carsetti, Isabella Quinti

**Affiliations:** 1Department of Molecular Medicine, Sapienza University of Rome, 00185 Rome, Italy; cinzia.milito@uniroma1.it (C.M.); andrea.palladino@uniroma1.it (A.P.); giulia.garzi@uniroma1.com (G.G.); dinapoli.1100150@studenti.uniroma1.it (G.D.N.); 2Internal Medicine 1, Ca’ Foncello University Hospital, AULSS2, 31100 Treviso, Italy; francesco.cine@gmail.com (F.C.); riccardo.scarpa@unipd.it (R.S.); marcello.rattazzi@unipd.it (M.R.); 3Department of Medicine, DIMED, University of Padova, 35128 Padova, Italy; 4Department of Translational Medical Sciences, University of Naples Federico II, 80131 Naples, Italy; a.punziano@studenti.unina.it (A.P.); g.lagnese93@gmail.com (G.L.); spadaro@unina.it (G.S.); 5Regional Reference Centre for Primary Immune Deficiencies, Azienda Ospedaliera Universitaria Policlinico Umberto I, 00161 Rome, Italy; annamaria.pesce2017@gmail.com (A.M.P.); federica.pulvirenti@gmail.com (F.P.); 6Diagnostic Immunology Research Unit, Multimodal Medicine Research Area, Bambino Gesù Children’s Hospital, IRCCS, Viale di San Paolo 15, 00146 Rome, Italy; rita.carsetti@opbg.net

**Keywords:** COVID-19, SARS-CoV-2, inborn errors of immunity, antibody deficiency, incidence, mortality rate, monoclonal antibodies

## Abstract

Patients with severely impaired antibody responses represent a group at-risk in the SARS-CoV-2 pandemic due to the lack of Spike-specific neutralizing antibodies. The main objective of this paper was to assess, by a longitudinal prospective study, COVID-19 infection and mortality rates, and disease severity in the first two years of the pandemic in a cohort of 471 Primary Antibody Defects adult patients. As secondary endpoints, we compared SARS-CoV-2 annual mortality rate to that observed over a 10-year follow-up in the same cohort, and we assessed the impact of interventions done in the second year, vaccination and anti-SARS-CoV-2 monoclonal antibodies administration on the disease outcome. Forty-one and 84 patients were infected during the first and the second year, respectively. Despite a higher infection and reinfection rate, and a higher COVID-19-related mortality rate compared to the Italian population, the pandemic did not modify the annual mortality rate for any cause in our cohort compared to that registered over the last ten years in the same cohort. PADs patients who died from COVID-19 had an underlying end-stage lung disease. We showed a beneficial effect of MoAbs administration on the likelihood of hospitalization and development of severe disease. In conclusion, COVID-19 did not cause excess mortality in Severe Antibody Deficiencies.

## 1. Introduction

Patients with Inborn Errors of Immunity (IEI) represent interesting models to provide insight on different immunological mechanisms involved in protection against infections. The protective role of functional innate immune responses has been recently shown in patients with IEI and Innate Immune Defects impairing type I interferon responses who developed a severe disease course [1]. Likewise, patients with IEI and severe antibody deficiencies (PADs) might help to better clarify the distinct role of antibodies in protection from SARS-CoV-2 infectivity and from COVID-19 severity [2,3,4].

Primary antibody deficiencies (PADs) are a heterogeneous group of immune disorders characterized by defective antibody production and/or inability to mount specific antibody responses. Patients with X-linked or Autosomal Recessive Agammaglobulinemia lack mature circulating B lymphocytes and present a severe decrease in all Ig subtypes, whereas patients with Common Variable Immunodeficiency (CVID) have dysfunctional B lymphocytes and decreased circulating levels of IgG, IgA and possibly IgM. In Hyper IgM Syndrome (HIGM), a defect may be present either in the T cell-dependent B cells’ co-stimulation or in the class switch recombination process, leading to decreased IgG and IgA levels with possible increase in circulating IgM. Good’s syndrome is a phenocopy of PAD occurring in patients with an underlying thymoma [5].

Neutralizing antibodies are widely considered the main protective mechanisms for SARS-CoV-2 infectivity, directly preventing the virus from replicating [6]. After SARS-CoV-2 immunization and infection, PADs patients showed low or absent neutralizing antibodies, generation of low frequency of specific memory B cells with low binding capacity to the Spike protein, absence of Receptor Binding Domain-positive memory B cells, and variable generation of Spike-specific T cells, indicating a different capability of B cells to undergo somatic mutation and affinity maturation in the germinal centers which are indispensable for the establishment of long-term immunity [7,8]. However, despite the functional impairment of their B cell compartment, when infected with SARS-CoV-2, one fourth of the PADs adult patients remained asymptomatic and half of them showed a mild disease [9,10]. The discrepancy between the clinical evidence and the impaired B cell function underlines the limitation to consider the specific antibody responses as a main correlate of clinical protection against severe forms of COVID-19 and death.

The main objective was to assess COVID-19 infection, and mortality rates and disease severity in the first two years of the SARS-CoV-2 pandemic in a cohort of Primary Antibody Defects adult patients. As secondary endpoints, we considered: (a) SARS-CoV-2 annual mortality rate in comparison to that observed over a 10-year follow-up in the same cohort; (b) the impact of vaccination and anti-SARS-CoV-2 monoclonal antibodies administration on the disease outcome.

## 2. Materials and Methods

### 2.1. Study Design

Since the beginning of the SARS-CoV-2 pandemic in Italy, we have conducted a still ongoing, longitudinal, prospective study in a cohort of 471 adult patients with Inborn Errors of Immunity (IEI) and severely impaired antibody responses affected with SARS-CoV-2 infection. The overall annual mortality rate observed in the period 1 March 2020 to 22 February 2022 was compared with the annual mortality rate observed in the previous ten years in the same cohort of patients followed at three Italian Referral Hospitals for adults with IEI (Padua, Northern Italy; Rome, Central Italy; Naples, Southern Italy). SARS-CoV-2 infection was diagnosed by reverse transcription-polymerase chain reaction (RT-PCR). All patients were tested every time they attended a hospital site, in an inpatient and outpatient setting, when a positive familiar contact was identified irrespective of symptoms, and upon onset of COVID-19-like illness. Easy access to the test had been granted since the very beginning of the pandemic, due to assumed at-risk status. Data regarding microbiological diagnosis of SARS-CoV-2 infection, disease severity, hospitalization, vaccination status, and SARS-CoV-2-specific treatments were derived from the existing database on PAD diagnosis and follow-up, already shared between the three centers participating in the Italian Primary Immune Deficiencies Network (IPINet) registry. Due to the status of regional referral centers, the three institutions had been directly involved in the management of each single infection, thus avoiding the risk of data loss. Data regarding the general Italian population were available at the Italian National Health Institute (Istituto Superiore di Sanità-ISS) website.

COVID-19 severity was defined according to WHO classification [11]. Re-infection data for comparison with the general population were recorded between 24 August 2021 and 2 March 2022, according to data availability from ISS reports. During the study time, patients continued their therapies, including immunoglobulin substitution as a standard therapy for the underlying antibody defect. From March 2021, according to the Italian vaccination rules, patients who agreed to undergo SARS-CoV-2 immunization received the BNT162b2 vaccine (Pfizer) administered as prescribed, in 2 doses, 21 days apart followed by a booster dose 3–4 months later. From March 2021, soon after diagnosis of COVID-19 disease, patients were treated with monoclonal antibodies unless they refused. In Italy, the original Wuhan strain was first isolated on 21 February 2020, and the number of active cases peaked on 21 March 2020. The SARS-CoV-2 B.1.1.7 (Alpha) variant was first isolated in December 2020. Three peaks of active cases were reported in April 2020, in November 2020, and in March 2021. During the last peak, the B.1.617.2 (Delta) variant was isolated for the first time and it is still circulating. On 28 November 2021, the B.1.1.529 Omicron variant was first identified in Italy. The new peak of the epidemic curve has been reached recently. 

The study was approved by the Ethical Committee of the Sapienza University of Rome (CE 5834, Prot. 0521/2020). The same cohort of adult patients with IEI and severely impaired antibody responses have been enrolled in the AIEOP/IPINET Italian registry that analyzes, retrospectively from 2010 to 2016 and prospectively since 2016, individual clinical and immunological data collected annually. The IPINet registry and the related informed consent forms have been approved by the local ethical committee (CE 4604, n. 316/2016). Both studies were performed in accordance with the Good Clinical Practice guidelines, the International Conference on Harmonization guidelines, and the most recent version of the Declaration of Helsinki.

### 2.2. Statistical Analysis of Numerical Data

Fisher’s exact test was used to investigate the infection and mortality rate in comparison to the Italian population, the effect of vaccination, and MoAbs on COVID-19 severity and mortality. Binomial logistic regression models were fitted to calculate odds ratios (OR) with 95% confidence intervals (CI) for the need of hospitalization and the presence of severe disease in association with MoAbs administration and vaccination status. Multivariable logistic regression analysis was then performed, to confirm the findings, taking into consideration age and sex as covariates. The impact of vaccination and MoAbs on disease severity and mortality was found by statistical significance which was considered as a two-tailed *p <* 0.05. All the analyses were performed using IBM SPSS statistics 27.0.

## 3. Results

### 3.1. Incidence/Infection Rate

The study included patients with Common Variable Immune Deficiency (n. 427), X-linked or Autosomal Recessive Agammaglobulinemia (n. 26), Good’s syndrome (n. 12), and Hyper IgM Syndrome (n. 6) diagnosed according to the ESID criteria (https://esid.org). In this cohort, since the beginning of the pandemic, the number of SARS-CoV-2 infections was 125/471 (26.54%); in detail, 112/427 (26.23%) patients with CVID; 9/26 (34.62%) with XLA or Agammaglobulinemia; 3/12 (25.00%) with Good’s syndrome, and 1/6 (16.67%) with HIGM syndrome were found infected. Cumulative infection rate was higher than that reported in the Italian population: (26.54% vs. 21.42%, *p* = 0.0076). Forty-one patients were infected within the first year (8.70% vs. 4.89% of the Italian population, *p* < 0.001), and 84 patients were infected in the second year (17.83% vs. 16.40 % of the Italian population, *p* = 0.439). It is possible to hypothesize that these infection rates were high but somehow mitigated by the application of the preventive measures our patients are used to following since the diagnosis of primary immune deficiency [12]. As expected, the rapid spread of the highly contagious Omicron variant [13] overcame these precautions, causing the recently observed increase in the number of positive cases (Figure 1).

### 3.2. Vaccination and Monoclonal Antibodies Treatment

In the second year of the pandemic, since 1 March 2021, we introduced two main strategies in the disease management based on: (1) SARS-CoV-2 vaccination by BNT162b2 (Pfizer–BioNtech), or mRNA-1273 (Moderna) administered according to the Italian national vaccination program: first dose and second dose were administered between March and April 2021, the booster dose between October and November 2021; (2) monoclonal antibodies (MoABs) administration available as treatment for SARS-CoV-2-infected fragile patients [14] (Figure 1). Between March 2021 and December 2021, 24 infections were identified before the appearance of the Omicron strain, whose pathogenicity in fragile patients is still unknown. Fourteen of these patients were infected before having the chance to be vaccinated and two patients refused vaccination due to personal concerns. Of the remaining eight patients, one became infected after one dose and seven were infected after two doses.

Since the spread of the Omicron variant in December 2021, we registered 60 additional infections, of which 54 were new infections and 6 re-infections: 46 patients were fully vaccinated, 11 had received two doses, and 3 did not undergo vaccination due to personal concerns.

Data on SARS-CoV-2 re-infection in the Italian population have been available from ISS since 24 August 2021. In the time interval between 24 August 2021 and 2 March 2022, the re-infection rate reported in the general population was 3%. A total of seven re-infection cases were observed in the same time interval in our PAD cohort: three in fully vaccinated patients, three in patients who underwent two doses of vaccine, and one in a patient who received one dose of vaccine only; six of seven were documented after the Omicron variant spread. Thus, the re-infection rate in the PAD cohort was 10%, higher when compared to the rate observed in the Italian population (*p* = 0.003). Six of the re-infections occurred in the CVID patient subgroup (10% vs. 3%, *p* < 0.001, when compared to the Italian population).

Since our data showed absent or low immune responses in our patients [7], we introduced the treatment with SARS-CoV-2 MoABs in recently infected patients (within 5 days from SARS-CoV-2 swab positivity by PCR). Therapeutic monoclonal antibodies used until December 2021 included imdevimab, casirivimab, etesivimab, regdanivimab, and bamlanivimab. These were used individually (bamlanivimab) or as combinations (casirivimab and etesivimab or bamlanivimab and imdevimab) to prevent viral resistance. Twelve patients received MoAbs: three patients had received two doses of vaccine, and nine patients were not immunized. Moderate/severe COVID-19 was diagnosed in 4/12 patients: two of them refused vaccination and two received two doses of vaccine, of which one died.

After the spread of the Omicron variant in December 2021, a total of 45 patients received Sotrovimab, a MoAb proved to be active against the variant [15]. Moderate/severe COVID-19 was diagnosed in 5/45 patients: three vaccinated with three doses, and two patients who had received two doses. None died.

### 3.3. COVID-19 Disease Severity and Fatality Rate

Table 1 summarizes COVID-19 severity stages, rate of hospitalization, and fatality rate in the first and in the second year.

The percentage of moderate/severe COVID-19 and hospitalizations decreased in the second year of the pandemic from 39.02% to 16.67% (*p* = 0.008). During the first two years of the pandemic, the COVID-19-related mortality rate in our cohort of PADs patients was higher when compared to the mortality registered in the Italian general population (4% vs. 1.19% *p* = 0.013). Two patients died in the first year, with a COVID-19 fatality rate accounting for 4.87%, and three patients died in the second year, with a COVID-19 fatality rate accounting for 3.57% (Table 1). The analysis of the CVID sub-population demonstrated that during the first year of pandemic, no differences were observed with the other PAD in COVID-19 severity, hospitalization, and mortality rate. Differently, during the second year, in the CVID cohort, we recorded lower incidence of hospital admission (due to moderate/severe COVID-19 stage), and severe COVID-19 course in comparison to the other PADs cohort (14.7% vs. 16.67%, *p* = 0.049 and 2.67% vs. 33.33%, *p* = 0.008, respectively) (Table 1).

The binomial logistic regression analysis performed in the 84 patients infected in the second year allowed us to assess the effects of vaccination status and MoAbs administration on the likelihood of admission to the hospital, development of severe disease, and death. COVID-19-related hospital admission and mortality rate were not significantly higher in non-vaccinated SARS-CoV-2-infected PAD in comparison to patients who underwent full immunization (24% vs. 15.5% and 4% vs. 3.4%, respectively) (Table 1). Differently, MoAbs administration resulted to be significantly associated with a decreased likelihood of admission to the hospital (*p* = 0.020, OR 0.253, 95%CI 0.079–0.808), with no impact on death and development of severe disease (Table 2). However, after adjustment for age, sex, and vaccination status, multivariable logistic regression analysis highlighted a beneficial effect of MoAbs administration both on the likelihood of hospitalization (*p* = 0.009; OR 0.187, 95%CI 0.053–0.653) and development of severe disease (*p* = 0.045; OR 0.095, 95%CI 0.009–0.951), thus confirming our previous data observed in a lower number of patients [14].

In this cohort, when considering CVID patients (*n* = 75) only, binomial logistic regression analysis, when adjusted for age, sex, and vaccination status, confirmed the positive impact of MoAbs administration on the risk of hospitalization (*p* = 0.029, OR = 0.199, 95%CI 0.047–0.844).

### 3.4. PADs Fatality Rate

Remarkably, three of five patients who died from COVID-19 over two years of pandemic had an underlying end-stage (defined as chronic oxygen dependent respiratory failure) Granulomatous Lymphocytic Interstitial Lung Disease (GLILD) (Table 3) [16]. Chronic lung disease remains a major clinical problem in PADs. Overall, after initiation of Ig replacement therapy, in PADs patients, we observed a reduction in the prevalence of pneumonia, whereas the cumulative risk of developing chronic lung disease increases in relation to age at diagnosis and diagnostic delay. The increased prevalence of chronic lung disease may be related both to the immune-mediated GLILD and to the well-known “vicious circle” infection-inflammation-remodeling, sustained by other immunological co-factors such as low frequency of memory B cells, very low IgA serum level (<7 mg/dL), and poor response to vaccination [16,17].

In the pandemic period, we counted an additional four and seven deaths for COVID-19-unrelated causes in the first year and in the second year of the pandemic, respectively (Table 4). In relation to the causes of death listed, when a known infectious agent was documented, (e.g., CMV) the cause of death was attributed to the infection, even in chronic lung disease patients. When chronic lung disease is reported as the cause of death, it stands for fatal exacerbation of an end-stage respiratory failure without a documented viral or bacterial or fungal pathogen. Of note, no previous infection, not even influenza, has been so efficiently investigated and accurately microbiologically documented on a large scale as SARS-CoV-2, both in the inpatient and outpatient setting. Thus, we cannot exclude that, at least in some of the listed cases, an unrecognized or just not specifically documented viral or bacterial infection may have at least triggered the fatal respiratory exacerbation.

As shown in Figure 2, the annual fatality rate for any cause registered, in our cohort, in the last 10 years, ranged from 1.06 to 2.34 per cent. The annual mortality rate in the first two years of the pandemic remained within this range (2020: 1.27 per cent; 2021: 2.13 per cent).

## 4. Discussion

In the first two years of the pandemic, the impaired specific antibodies production accounted for the high infection susceptibility to SARS-CoV-2 in PADs patients, confirming the main role of specific antibodies in the protection against SARS-CoV-2 infection. Our study demonstrates that, despite a high infection rate, a high reinfection rate, and a high COVID-19-related mortality rate, the pandemic did not modify the annual mortality rate for any cause in a population of adult patients with PADs. In fact, in the study time, the latter was similar to the mortality rate observed in the last 10 years in the same cohort, and PADS patients who died from COVID-19 over the two years had an underlying severe lung disease. Despite the percentage of moderate/severe COVID-19 and hospitalizations decreased in the second year of the pandemic after changing the patients’ management by vaccination and by introducing monoclonal antibody therapies; in SARS-CoV-2- infected patients, we could not prove a beneficial effect of full immunization on COVID-19 mortality nor on hospitalization, while a protective effect was evident from the reduced number of hospitalized patients when treated with Spike-specific MoAbs. The low or even absent antibody response after infection and/or immunization, generated considerable anxiety in our patient population who were aware of their immunodeficiency. The shift of attention from an immunity exclusively mediated by antibodies to a more comprehensive model of protection mediated by immune cells might help in the doctor–patient communication. The high number of patients who had a mild disease and who survived COVID-19 offers the opportunity to discuss the protective role of immune cells, other than antibodies and B cells, in COVID-19 disease. Specific functional and protective virus-specific T cell memory generated [18] could play a critical role in mitigating the disease. Specific T cell activity has been widely recognized as a main protective mechanism in COVID-19 [19], and it recently gained more attention as SARS-CoV-2 variants emerged [20]. All patients with Agammaglobulinemia and the majority of patients with Common Variable Immune Deficiency have functional T cell-dependent immunity, as also demonstrated by their ability to respond to stimulation with influenza virus antigens [21]. SARS-CoV-2 specific T cells residing in the lung or in the upper respiratory tract may contribute to control of infection, similarly to T cells resident in the lung locally controlling other respiratory tract viral infections [22]. Moreover, T cell activity and humoral immunity against SARS-CoV-2 proteins has been shown in PADs patients convalescent from COVID-19 [23]. Of note, while the natural course of COVID-19 is controlled by the function of the innate immune system, with a secondary involvement of T and B cells, SARS-CoV-2 vaccines are designed to force the adaptive immune system. Differently from infection, poor Spike-specific T cell responses were generated in PADs patients by the first two doses of immunization [24]. In PADs, the first antigenic stimuli provided by vaccination with a novel virus with Spike- and the RBD-domains differing from the S proteins of most members of the family of coronavirus [8,17], and may have not been sufficient to induce antibody responses and to generate memory B cells as well as to induce effective T cells, and to activate circulating T follicular helper cells [2]. Differently, additional booster immunizations could lead to enhancement of SARS-CoV-2 circulating T follicular helper cells as in vaccination against influenza [21], and vaccine booster doses might render patients’ antigen-specific T cells more responsive to stimulation or increase their numbers. Interestingly, our patients have similar immunological conditions to the low-responder convalescents described in a recent paper [25] who did not report any relevant symptoms during SARS-CoV-2 infection. Apart from T cell contribution, the lack of induction of adaptive B cell immunity might also be compensated by functional innate immunity cells [26,27]. Only in patients with innate immune deficiencies, indeed, a more severe COVID-19 course and a poor prognosis have been suggested, due to a pivotal role of innate immunity in the response against SARS-CoV-2 infection [1,28]. In this way, polyclonal immunoglobulin substitutive treatment, regularly administered in all our patients, could also have contributed to mitigate COVID-19 due to their immunomodulatory function, mainly active on innate immune cells [29], even when administered at replacement dosages.

A limitation of our study was the low rate of re-infection reported in our cohort possibly related to the study observational period, because in the general population and in PADs, the number of re-infection dramatically increased just at the end of the study period, due to the spread of the first Omicron variant in Italy. However, this should not impair our conclusions in terms of mortality and therapeutic management. This could just open the way for specifically designed follow-up studies. A further limitation was the impossibility to perform a proper timely comparison between PADs patients and the general populations. The vaccination rate registered in our cohort at the end of the study period was > 95%, whereas the vaccination rate reported in the Italian population was 86.04% (2 March 2022). However, according to the Italian national vaccination program, the first and the second dose of BNT162b2 (Pfizer–BioNtech), or mRNA-1273 (Moderna) COVID-19 vaccine were administered to PADs and other at-risk patients between March and April 2021 whereas the booster dose between October and November 2021. Vaccination was offered, without limitation, to the general adult Italian population only from July 2021.

## 5. Conclusions

The identification of a strategy to control SARS-CoV-2 infection and to mitigate COVID-19 severity in patients with impaired immune mechanisms of protection from infections remains a priority. The administration of MoAbs resulted to be significantly associated with a decreased hospitalization, as well as substitutive therapy with polyclonal gamma globulins might further contribute to mitigate the clinical consequences of COVID-19. The potential role of therapy with polyclonal gamma globulins is based not only on their replacement and immunomodulatory effects, but also on the possibility to infuse SARS-CoV-2 antibodies by the coming of new lots of gamma globulins [30]. Currently, on the basis of our results, we are continuing to administer MoAbs, with a proven efficacy against the new viral variants, to immunize with additional booster doses of vaccine, to analyze the short- and long-term immune responses, and to administer newly available antiviral drugs. An additional strategy in the management of COVID-19 disease in PADs patients could be based on the prophylactic use of MoAbs, which might modify the clinical course of the disease. Finally, accurate and prompt microbiological diagnosis and treatment as well as characterization of the immunological determinants of response to vaccination and infections has always been a cornerstone of the management of PADs patients. Thus, understanding COVID-19 clinical and biological behavior in patients with IEI will hopefully help in designing new therapeutic strategies for immunocompetent populations.

## Figures and Tables

**Figure 1 biomedicines-10-01026-f001:**
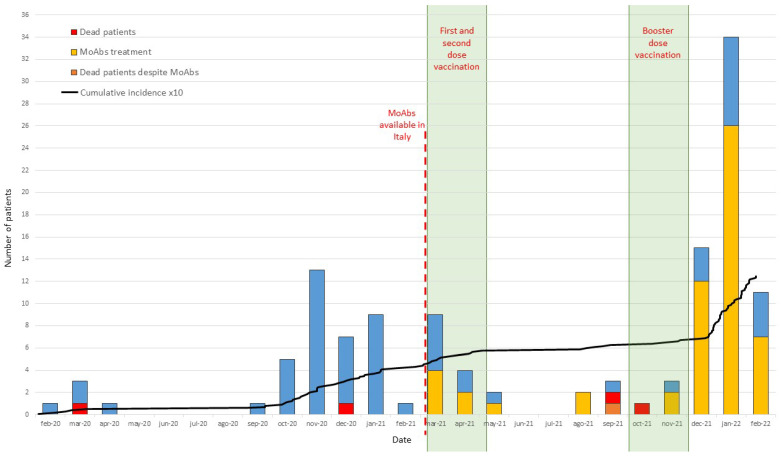
Number of SARS-CoV-2-infected patients in the cohort of IEI patients with antibody deficiencies in the first twenty-four months of the pandemic. In blue, patients who recovered; in yellow, patients treated with MoAbs and recovered; in orange, patients who died despite MoAbs; in red, patients who died. Cumulative incidence is indicated by a black line, and initial date of MoAbs availability in Italy is indicated by a red dotted line. The two periods of first/second doses of vaccine administration, and of booster dose administration are indicated by a shadowed area.

**Figure 2 biomedicines-10-01026-f002:**
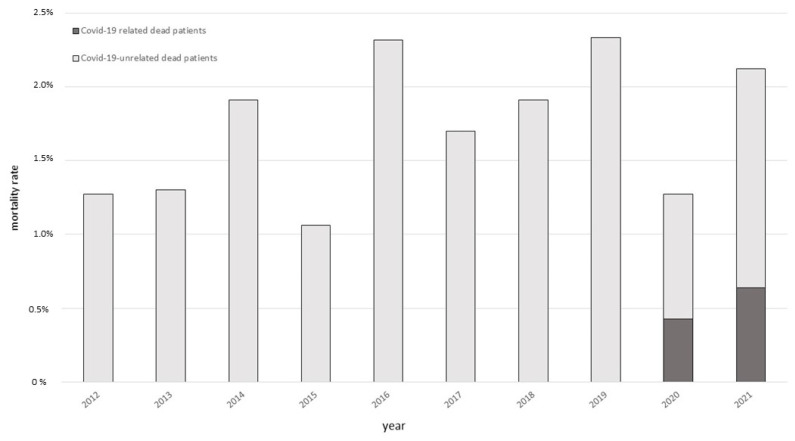
Annual mortality in the cohort of IEI patients with Primary Antibody Deficiencies in the period 2012–2021. Dead patients for COVID-19-related causes are indicated in gray; dead patients for COVID-19-unrelated causes are indicated in black.

**Table 1 biomedicines-10-01026-t001:** Summary of absolute number and percentages of SARS-CoV-2 infections in PADs patients grouped according to COVID-19 stages. Hospitalizations and deaths observed in the first and second year of the cohort study were also reported.

		Asymptomatic/Mild *n* (%)	Moderate/Severe *n* (%)	Hospitalizations *n* (%)	Deaths *n* (%)
**1st YEAR** **(*n* = 41)**	All PADs	25 (60.98%)	16 (39.02%)	16 (39.02%)	2 (4.87%)
	CVID only	22 (59.5%)	15 (40.5%)	15 (40.5%)	2 (5.4%)
**2nd YEAR** **(*n* = 84)**	All PADs	70 (83.33%)	14 (16.67%)	14 (16.67%)	3 (3.57%)
	CVID only	64 (85.3%)	11 (14.7%)	11 (14.7%)	2 (2.67%)

**Table 2 biomedicines-10-01026-t002:** Impact of Vaccination and MoAbs administration on COVID-19 course during the second year of pandemic (84 SARS-CoV-2-infected PADs patients).

Hospitalization	*n* (%)	Unadjusted OR (95% CI)		Adjusted OR (95% CI)	
Vaccinated vs. not vaccinated	9 (15.5) vs. 6 (24.0)	0.582 (0.182–1.857)	*p* = 0.360 *	0.453 (0.131–1.561)	*p* = 0.210 §
MoAbs vs. not MoAbs	6 (10.7) vs. 9 (32.1)	0.253 (0.079–0.808)	*p* = 0.020 *	0.187 (0.053–0.653)	*p* = 0.009 #
**Severe Disease**	** *n* ** **(%)**	**Unadjusted** OR (95% CI)		**Adjusted** OR (95% CI)	
Vaccinated vs. not vaccinated	3 (5.2) vs. 2 (8.0)	0.627 (0.098–4.006)	*p* = 0.622 *	0.664 (0.096–4.564)	*p* = 0.677 §
MoAbs vs. not MoAbs	1 (1.8) vs. 4 (14.3)	0.109 (0.012–1.028)	*p* = 0.053 *	0.095 (0.009–0.951)	*p* = 0.045 #
**Mortality**	** *n* ** **(%)**	**Unadjusted** OR (95% CI)		**Adjusted** OR (95% CI)	
Vaccinated vs. not vaccinated	2 (3.4) vs. 1 (4)	0.857 (0.074–9.909)	*p* = 0.902 *	0.781 (0.059–10.294)	*p* = 0.851 §
MoAbs vs. not MoAbs	1 (81.8) vs. 2 (7.1)	0.236 (0.020–2.726)	*p* = 0.248 *	0.110 (0.013–2.147)	*p* = 0.167 #

* Binomial logistic regression analysis. § Adjusted for age and gender by multivariable logistic regression analysis. # Adjusted for age, gender, and vaccination status by multivariable logistic regression analysis.

**Table 3 biomedicines-10-01026-t003:** Data of patients who died during the two years of SARS-CoV-2 pandemic.

Patient	Sex	Age	PID	Comorbidity	Date of Infection	Vaccination Status	MoABs Therapy
1	F	59	CVID	GLILD, chronic respiratory failure	March 2020	Not done	No
2	M	52	CVID	GLILD, bilateral lung transplantation, chronic respiratory failure	December 2020	Not done	No
3	F	48	CVID	GLILD, chronic respiratory failure	September 2021	2 doses	Yes
4	M	78	CVID	Chronic heart failure, bronchiectasis	September 2021	2 doses	No
5	M	46	XLA	Post-poliomyelitis flaccid paralysis. Chronic obstructive pulmonary disease	October 2021	Refused	No

**Table 4 biomedicines-10-01026-t004:** Number of patients dead for a given cause observed in the last 10 years in the cohort of Italian PADs patients.

Year	All Causes of Death
**2012**	4, cancer; 1, CMV disseminated infection; 1, autoimmune cytopenias
**2013**	2, cancer; 1, chronic lung disease *; 1, CMV disseminated infection; 2, enteropathy
**2014**	4, cancer; 3, chronic lung disease *; 1, autoimmune cytopenias; 1, enteropathy
**2015**	2, cancer; 2, chronic lung disease *, 1, enteropathy
**2016**	5, cancer; 3, autoimmune cytopenias; 2, chronic lung disease *; 1, hepatic disease
**2017**	2, cancer; 2, autoimmune cytopenias; 1, chronic lung disease *; 3, enteropathy
**2018**	3, cancer; 3, autoimmune cytopenias; 2, chronic lung disease *; 1, hepatic disease
**2019**	3, cancer; 3, hepatic diseases; 3, chronic lung disease *; 1, autoimmune cytopenias; 1 enteropathy
**2020**	3, chronic lung disease *; 1, hepatic disease; 2, COVID-19
**2021**	2, cancer; 3, chronic lung disease *; 1, meningitis; 1, autoimmune cytopenias; 3, COVID-19

* chronic lung disease stands for fatal exacerbation of end-stage respiratory failure in chronic lung disease, without a definite microbiological isolation.

## Data Availability

All data necessary to support the reported results are present in the main text of the article.

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
