# Peer review of "Mortality in Severe Antibody Deficiencies Patients during the First Two Years of the COVID-19 Pandemic: Vaccination and Monoclonal Antibodies Efficacy"

_biomedicines, 2022, doi:10.3390/biomedicines10051026_

Round 1

Reviewer 1 Report

The study by Milito et al. studies incidence and severity of COVID-19 infections in patients with primary immunodeficiency diseases, namely common variable immune deficiency (n=427), X-linked or autosomal regressive Agammaglobulinemia (n=26), Good’s Syndrome (n=12), Hyper IgM Syndrome (n=6). It is surprising that neither the infection rate was much higher than in the Italian population nor mortality was much higher.

It might be useful to add a brief characterization of the pathology of the PAD, which are relevant for the study.

Was the difference between infection rate in the patient collective compared to the general Italian population similar in the first and second period of observation?

It is not surprising that chronic lung disease is a risk factor for death from COVID-19. Patients with PAD died from chronic lung disease also in the years prior to the SARS-CoV-2 pandemic. Did other respiratory infections (bacterial or influenza A virus) cause this mortality?

CVID is the most common PAD and expected to present a more homogenous population than all PAD patients together. It might be worth to analyse these cases separately and compare the data with those from the entire PAD patient group.

Was it verified that CVID patients had, as expected, low IgA levels?

Was the vaccination rate of the patients similar to the Italian population?

It appears that the numbers of re-infection are too low to draw valid conclusions.

The authors mentioned that not only antibody production was missing in the PAD cohort but also the T-cell response to the spike protein was low. What would be their hypothesis for the close to normal protection against COVID-19?

Minor

Table 2: inconsistent capitalization of SARS-CoV-2

Author Response

Q: The study by Milito et al. studies incidence and severity of COVID-19 infections in patients with primary immunodeficiency diseases, namely common variable immune deficiency (n=427), X-linked or autosomal recessive Agammaglobulinemia (n=26), Good’s Syndrome (n=12), Hyper IgM Syndrome (n=6). It is surprising that neither the infection rate was much higher than in the Italian population nor mortality was much higher. Was the difference between infection rate in the patient collective compared to the general Italian population similar in the first and second period of observation?

R: Thanks to Reviewer for her/his suggestions. We have extensively revised the manuscript according to the comments.

The comparison with the infection rate of the general Italian population during the first and second year has been included in paragraphs 3.1 and 3.2.

Q: It might be useful to add a brief characterization of the pathology of the PAD, which are relevant for the study.

R: We added in the introduction a brief characterization on the main PAD-associated immune alterations.

 Q: It is not surprising that chronic lung disease is a risk factor for death from COVID-19. Patients with PAD died from chronic lung disease also in the years prior to the SARS-CoV-2 pandemic. Did other respiratory infections (bacterial or influenza A virus) cause this mortality?

R: Thanks to Reviewer for giving us the opportunity to specify. “Chronic lung disease remains a major clinical problem in PADs.Overall, after initiation of Ig replacement therapy we observed a reduction in the prevalence of pneumonia, whereas the cumulative risk of developing chronic lung disease increases in relation to age at diagnosis and diagnostic delay. The increased prevalence of chronic lung disease is probably related to the well-known “vicious circle” infection-inflammation-remodeling, and related to other aspects such as low frequency of memory B cells, very low IgA serum level (<7 mg/dl) and poor response to vaccination (Ref. 16, 17). In relation to the causes of death listed in Table 3, when a known infectious agent was documented, (e.g. CMV) the cause of death was attributed to the infection, even in chronic lung disease patients. When chronic lung disease is reported as the cause of death, it stands for fatal exacerbation of an end stage respiratory failure without a documented viral or bacterial or fungal pathogen. Of note, no previous infection, not even influenza, has been so efficiently investigated and accurately microbiologically documented on a large scale as SARS-CoV-2, both in the inpatient and outpatient setting. Thus, we cannot exclude that, at least in some cases, an unrecognized or just not specifically documented viral or bacterial infection may have at least triggered the fatal respiratory exacerbation. Accordingly, we slightly modified the table caption.

Q: CVID is the most common PAD and expected to present a more homogenous population than all PAD patients together. It might be worth to analyse these cases separately and compare the data with those from the entire PAD patient group.

R: Thanks. We modified the analysis, by considering the CVID population separately from patients affected with other PADs and included these data in paragraph 3.3, and in Table 1.

Q: Was it verified that CVID patients had, as expected, low IgA levels?

R: According to ESID REGISTRY diagnostic criteria (https://esid.org), all CVID patients included in the study had low or undetectable IgA levels.

Q: Was the vaccination rate of the patients similar to the Italian population?

R: Thanks for the interesting question. We insert in the discussion: "The vaccination rate registered in our cohort at the end of the study period was >95%, whereas the vaccination rate reported in the Italian population was 86.04% (March 2, 2022). However, according to the Italian national vaccination programme, the first and the second dose of BNT162b2 (Pfizer–BioNtech), or mRNA-1273 (Moderna) COVID-19 vaccine were administered to PADs and other at-risk patients between March and April 2021 whereas the booster dose between October and November 2021. Vaccination was instead offered, without limitation, to the general adult Italian population only from July 2021. For this reason, it was not possible to perform a proper timely comparison”.

Q: It appears that the numbers of re-infection are too low to draw valid conclusions.

R: Thanks, we agree with this comment. The comparison of re-infection rate between PADs and general population is included in paragraph 3.2, where we have now added the results of the statistical analysis. We added in the conclusions: “The low rate of re-infection reported in our cohort could be related to the study observational period. In Italy, in the general population and in PAD patients, the number of re-infection dramatically increased just at the end of the study period due to the spread of the first Omicron variant. However, this should not impair our conclusions in terms of mortality and therapeutic management. This could just open the way for specifically designed follow-up studies”.

Q: The authors mentioned that not only antibody production was missing in the PAD cohort but also the T-cell response to the spike protein was low. What would be their hypothesis for the close to normal protection against COVID-19?

R: Thanks to offer the opportunity to discuss this point.

Minor

Q: Table 2: inconsistent capitalization of SARS-CoV-2.

R: Thanks to REV1 for her/his suggestion. Accordingly, we modified the text.

Reviewer 2 Report

  1. The research topic is important, however, some revisions are needed:

    Please clearly define the study aim.
    The introduction section should provide data on access to the COVID-19 test and all the procedures related to the Primary Antibody Defects in adult patients
    More precise data on study settings are needed. Which facilities were included, regional data, location, etc? This is particularly important for international readers.
    In the results section - simple statistics are applied which is a limitation of this study
    In the discussion section - please clearly present the limitations as well as practical implications of this study
    In the conclusions section - please remove "link to FDA website" from the conclusions. This section should be revised and based on its own findings.
    The importance of this study for international readers should be emphasized

Author Response

The research topic is important, however, some revisions are needed

R: Thanks to Reviewer for her/his suggestions. We have extensively revised the manuscript according to the comments.

Q: Please clearly define the study aim.

R: We thank the Reviewer for her/his kind suggestion. Accordingly, we modified the introduction by adding a final “aim of the study” paragraph.

Q: The introduction section should provide data on access to the COVID-19 test and all the procedures related to the Primary Antibody Defects in adult patients.

R: Thanks for the suggestion. We had already specified in the methods section some information regarding the access to the COVID-19 test and procedures related to the PADs adult patients. We have improved the description in the same section. It is interesting to note that no previous infection, in history, has been so efficiently investigated and accurately microbiologically documented as SARS-CoV-2 one, both in the inpatient and outpatient setting. The accuracy of this diagnostic approach has been particularly high, since the beginning of the pandemic, in the assumed at-risk populations, including PADs patients.

Q: More precise data on study settings are needed. Which facilities were included, regional data, location, etc? This is particularly important for international readers.

R: Thank you. We specified this issue in the Materials and Methods -  Study design section”

Q: In the results section - simple statistics are applied which is a limitation of this study

R: Thank you. We agree and now, in the Results section, we improved our statistical analysis by adding a comparison between CVID and other PADs, as suggested also by Reviewer 1. We added a binomial and a multivariate logistic regression analysis for the confirmation of the impact of vaccination and MoAbs administration on the COVID-19 course.

Q: In the discussion section - please clearly present the limitations as well as practical implications of this study

R: We inserted the limitations and the practical implications in the discussion section and in the conclusions. Main limitations: A limitation of our study was the low rate of re-infection reported in our cohort possibly related to the study observational period because in the general population and in PADs the number of re-infection dramatically increased just at the end of the study period, due to the spread of the first Omicron variant in Italy. However, this should not impair our conclusions in terms of mortality and therapeutic management. This could just open the way for specifically designed follow-up studies. A further limitation was the impossibility to perform a proper timely comparison between PADs patients and the general populations. The vaccination rate registered in our cohort at the end of the study period was > 95%, whereas the vaccination rate reported in the Italian population was 86.04% (March 2, 2022). However, according to the Italian national vaccination programme, the first and the second dose of BNT162b2 (Pfizer–BioNtech), or mRNA-1273 (Moderna) COVID-19 vaccine were administered to PADs and other at-risk patients between March and April 2021 whereas the booster dose between October and November 2021. Vaccination was instead offered, without limitation, to the general adult Italian population only from July 2021.

Main practical conclusion: “Currently, on the basis of our results, we are continuing to administer MoAbs, with a proven efficacy against the new viral variants, to immunize with additional booster doses of vaccine, to analyze the short- and long-term immune responses, and to administer newly available antiviral drugs. An additional strategy in the management of Covid-19 disease in PADs patients, could be based on the prophylactic use of MoAbs, that might modify the clinical course of the disease. Finally, accurate and prompt microbiological diagnosis and treatment as well as characterization of the immunological determinants of response to vaccination and infections has always been a cornerstone of the management of PADs patients. Thus, understanding COVID-19 clinical and biological behavior in patients with IEI will hopefully help designing new therapeutic strategies for immunocompetent populations.

Q: In the conclusions section - please remove "link to FDA website" from the conclusions. This section should be revised and based on its own findings.

R: We removed "link to FDA website" from the conclusions and included our findings.

Q: The importance of this study for international readers should be emphasized.

R: “We thank the Reviewer for her/his kind suggestion. We commented in the discussion and conclusions sections.

Round 2

Reviewer 1 Report

The authors have addressed my comments and I recommend the manuscript for publication.